# Intra-Host Evolution Analyses in an Immunosuppressed Patient Supports SARS-CoV-2 Viral Reservoir Hypothesis

**DOI:** 10.3390/v16030342

**Published:** 2024-02-23

**Authors:** Dominique Fournelle, Fatima Mostefai, Elsa Brunet-Ratnasingham, Raphaël Poujol, Jean-Christophe Grenier, José Héctor Gálvez, Amélie Pagliuzza, Inès Levade, Sandrine Moreira, Mehdi Benlarbi, Guillaume Beaudoin-Bussières, Gabrielle Gendron-Lepage, Catherine Bourassa, Alexandra Tauzin, Simon Grandjean Lapierre, Nicolas Chomont, Andrés Finzi, Daniel E. Kaufmann, Morgan Craig, Julie G. Hussin

**Affiliations:** 1Research Centre Montreal Heart Institute, Montréal, QC H1T 1C8, Canada; dominique.fournelle@umontreal.ca (D.F.); fatima.mostefai@umontreal.ca (F.M.); raphael.poujol@gmail.com (R.P.); jean.christophe.grenier@gmail.com (J.-C.G.); 2Département de Biochimie et Médecine Moléculaire, Université de Montréal, Montréal, QC H3C 3J7, Canada; 3Centre de Recherche du Centre Hospitalier de l’Université de Montréal (CRCHUM), Montréal, QC H2X 0A9, Canada; elsa.brunet-ratnasingham@umontreal.ca (E.B.-R.); amelie.pagliuzza.chum@ssss.gouv.qc.ca (A.P.); mehdi.benlarbi@umontreal.ca (M.B.); guillaume.beaudoin-bussieres@umontreal.ca (G.B.-B.); gabrielle.gendron-lepage.chum@ssss.gouv.qc.ca (G.G.-L.); catherine.bourassa.chum@ssss.gouv.qc.ca (C.B.); alexandra.tauzin@umontreal.ca (A.T.); simon.grandjean.lapierre@umontreal.ca (S.G.L.); nicolas.chomont@umontreal.ca (N.C.); daniel.kaufmann@chuv.ch (D.E.K.); 4Département de Microbiologie, Infectiologie et Immunologie, Université de Montréal, Montréal, QC H3C 3J7, Canada; 5Canadian Centre for Computational Genomics, Montréal, QC H3A 0G1, Canada; jose.hector.galvez@computationalgenomics.ca; 6Laboratoire de Santé Publique du Québec, Institut National de Santé Publique du Québec, Sainte-Anne-de-Bellevue, QC H9X 3R5, Canada; ines.levade@inspq.qc.ca (I.L.);; 7Centre Hospitalier de l’Université de Montréal (CHUM), Montréal, QC H2X 0C1, Canada; 8Division of Infectious Diseases, Department of Medicine, University Hospital and University of Lausanne, CH-1015 Lausanne, Switzerland; 9Research Centre, Centre Hospitalier UniversitaireSainte-Justine, Montréal, QC H3T 1C5, Canada; morgan.craig@umontreal.ca; 10Département de Mathématiques et de Statistique, Université de Montréal, Montréal, QC H3T 1J4, Canada; 11Département de Médecine, Université de Montréal, Montréal, QC H3C 3J7, Canada; 12Mila-Quebec AI Institute, Montréal, QC H2S 3H1, Canada

**Keywords:** SARS-CoV-2, immune escape mutations, viral genomics, phylogenetics

## Abstract

Throughout the SARS-CoV-2 pandemic, several variants of concern (VOCs) have been identified, many of which share recurrent mutations in the spike glycoprotein’s receptor-binding domain (RBD). This region coincides with known epitopes and can therefore have an impact on immune escape. Protracted infections in immunosuppressed patients have been hypothesized to lead to an enrichment of such mutations and therefore drive evolution towards VOCs. Here, we present the case of an immunosuppressed patient that developed distinct populations with immune escape mutations throughout the course of their infection. Notably, by investigating the co-occurrence of substitutions on individual sequencing reads in the RBD, we found quasispecies harboring mutations that confer resistance to known monoclonal antibodies (mAbs) such as S:E484K and S:E484A. These mutations were acquired without the patient being treated with mAbs nor convalescent sera and without them developing a detectable immune response to the virus. We also provide additional evidence for a viral reservoir based on intra-host phylogenetics, which led to a viral substrain that evolved elsewhere in the patient’s body, colonizing their upper respiratory tract (URT). The presence of SARS-CoV-2 viral reservoirs can shed light on protracted infections interspersed with periods where the virus is undetectable, and potential explanations for long-COVID cases.

## 1. Introduction

Several SARS-CoV-2 variants of concern (VOCs) have convergent mutations in the spike glycoprotein’s receptor-binding domain (RBD) that coincide with known epitopes [1]. Mutations in this genomic region affect the ability of the spike (S) to enter the cell via the ACE2 receptor and have been linked with higher transmission rates and/or immune escape [2,3].

While in most cases, SARS-CoV-2 infections are cleared within a few days, key mutations develop de novo during prolonged infections in patients with immunosuppressive conditions. These infections can last for several months, and their viral mutation rate is higher than in shorter infections in immunocompetent patients [4]. For this reason, it is suspected that protracted infections are one of the drivers of SARS-CoV-2’s genomic evolution and a source of immune escape variants [5]. One such example is S:E484K that was found in the former VOCs Beta and Gamma [6]. This mutation has been shown to result in virus immune escape properties such as resistance to anti-viral monoclonal antibodies (mAbs) and convalescent sera as well as reinfection [7].

It is known that viruses can hide in protected regions in the host’s body in viral reservoirs [8]. These viruses then linger and have the potential to re-emerge, thereby sustaining a protracted infection. Recent evidence shows that SARS-CoV-2 can be found in different anatomical locations months after the infection has been cleared from the respiratory system [9]. It has also been shown that different sets of viral mutations appear across tissues, meaning intra-host viral evolution can occur in other places than the upper respiratory tract (URT) [10]. These viral populations in untested tissues can evolve separately and migrate to the URT [11]. Unfortunately, most of the available SARS-CoV-2 genomes sequences come from samples taken in the URT, giving a potentially incomplete snapshot of the viral diversity present within the host.

Because there is more opportunity for viral evolution and the collection of samples in protracted infections of immunosuppressed individuals, these cases can be useful in understanding the dynamics of viral reservoirs. Recent studies have begun to fill this gap, providing longitudinal datasets of sequences collected from these individuals at different time points during their infections [11,12,13]. These datasets can give insights into evolutionary events that are not observed in acute infections, such as an instance of recombination between two viral strains [14] or the presence of distinct viral populations with immune escaping mutations in a single sample [15].

Here, we describe the genetic events that arose in a patient with non-Hodgkin lymphoma who was infected with SARS-CoV-2 for several months. Samples from this patient were collected by the Public Health Laboratory of Québec (LSPQ) in Canada. Through the phylogenetic and intra-host single nucleotide variant (iSNV) analysis of viral sequences, we found several viral populations containing immune escape mutations in the spike’s RBD. Further, we provide evidence of a mutational pattern suggestive of a viral reservoir that produced viruses capable of reinfecting the URT in this immunosuppressed patient.

## 2. Materials

### 2.1. Plasma and Antibodies

Plasma was collected, heat-inactivated for 1 h at 56 °C, and stored at −80 °C until use. Plasma from uninfected donors collected before the pandemic were used as negative controls and to calculate the seropositivity threshold in our ELISA, ADCC, and flow cytometry assays. The RBD-specific CR3022 was used as positive control in our ELISA assays, and the conformationally independent S2-specific CV3-25 mAb was used in our flow cytometry assays. Both antibodies were previously described [16,17]. Horseradish peroxidase (HRP)-conjugated Abs able to detect all Ig isotypes (anti-human IgM + IgG + IgA; Jackson ImmunoResearch Laboratories, West Grove, PA, USA) or ones specific to the Fc region of human IgG (Invitrogen, Waltham, MA, USA), IgM (Jackson ImmunoResearch Laboratories, West Grove, PA, USA), or IgA (Jackson ImmunoResearch Laboratories, West Grove, PA, USA) were used as the secondary Abs in the ELISA assays. Alexa Fluor-647-conjugated goat anti-human Abs able to detect all Ig isotypes (anti-human IgM + IgG + IgA; Jackson ImmunoResearch Laboratories, West Grove, PA, USA) were used as the secondary Ab in the flow cytometry experiments.

### 2.2. Cell Lines

293T human embryonic kidney cells (obtained from ATCC) were maintained in Dulbecco’s modified Eagle’s medium (DMEM) (Wisent, Saint-Jean-Baptiste, CA, USA) containing 5% fetal bovine serum (FBS) (VWR, Mont-Royal, CA, USA) and 100 mg/mL penicillin–streptomycin (Wisent, Saint-Jean-Baptiste, CA). CEM.NKr CCR5+ cells (NIH AIDS reagent program) were maintained in Roswell Park Memorial Institute (RPMI) 1640 medium (GIBCO) containing 10% FBS and 100 mg/mL of penicillin–streptomycin. The 293T-ACE2 cell line was previously reported [18]. CEM.NKr CCR5+ cells stably expressing the SARS-CoV-2 S glycoproteins were previously reported [19]. All cell lines were maintained at 37 °C under 5% CO_2_.

### 2.3. Plasmids

The plasmids expressing the human coronavirus S glycoprotein of D614G and B.1.160 (carrying the mutations S477N and D614G) were previously reported [20].

## 3. Methods

### 3.1. Viral Databases

SARS-CoV-2 consensus sequence data were obtained from the LSPQ database through the CoVSeQ consortium: https://covseq.ca (accessed on 16 November 2021). Only sequences that were covered at more than 90% and a mean depth of 50× for Illumina and 16× for Oxford Nanopore technologies (ONT) with no previously documented frameshift, less than 5% N, at most 5 ambiguous bases, were used. Serial sequences from two patients described by Lee et al. [10] (P1 and P2) were obtained from NCBI’s Sequence Read Archive (study SRP357108). One of them (P1) did not have iSNV in the spike’s RBD and was excluded from this study. A total of 147,537 representative SARS-CoV-2 Illumina libraries from 2020 and 2021 were downloaded from NCBI and served as a reference dataset to compare patient data (see *Intra-Host Analysis* below). Metadata for the patient were obtained as part of BQC-19 (PMID: 34010280).

### 3.2. Whole-Genome Sequencing and Consensus Sequence Generation

All LSPQ sequencing data were analyzed using the GenPipes [21] Covseq pipelines to produce variant calls and consensus sequences. Samples were sequenced on Illumina or ONT. Regardless of the sequencing technology, data were initially processed to remove any host sequences by aligning them to a hybrid human (GRCh38) and SARS-CoV-2 (NCBI accession number MN908947.3) reference. Any sequences that aligned to the human portion of the hybrid reference were removed from downstream analysis. For Illumina sequencing data, raw reads were first trimmed using cutadapt (v2.10) available online: https://cutadapt.readthedocs.io/en/v2.10/installation.html (accessed on 21 January 2024) and then aligned to the reference using bwa-mem (v0.7.17) available online: https://bio-bwa.sourceforge.net/ (accessed on 21 January 2024). Aligned reads were filtered using sambamba (v0.7.0) available online: https://lomereiter.github.io/sambamba/ (accessed on 21 January 2024) to remove paired reads with an insert size outside the 60–300 bp range, unmapped reads, and all secondary alignments. Then, any remaining ARTIC primers (v3) available online: https://github.com/artic-network/primer-schemes/tree/master/nCoV-2019 (accessed on 21 January 2024) were trimmed with iVar (v.1.3.4) available online: https://andersen-lab.github.io/ivar/html/manualpage.html (accessed on 21 January 2024). To create a consensus representative of the most abundant species in the sample, a pileup was produced using Samtools (v1.9) available online: https://www.htslib.org/doc/ (accessed on 21 January 2024) which was used as an input for FreeBayes (v1.2.2) available online: https://github.com/freebayes/freebayes (accessed on 21 January 2024). For ONT sequencing data, raw signals were basecalled using guppy (v3.4.4) available online: https://nanoporetech.com/support (accessed on 21 January 2024) with the High-Accuracy Model (dna_r9.4.1_450bps_hac) available online: https://esr-nz.github.io/gpu_basecalling_testing/gpu_benchmarking.html (accessed on 21 January 2024). Reads outside the expected size range (400–700 bp) were removed from the analysis. Reads were then aligned to the reference using minimap2 (v.2.17) available online: https://github.com/lh3/minimap2/releases/tag/v2.26 (accessed on 21 January 2024) and filtered to remove incorrect primer pairs and randomly downsampled to keep 800× depth per strand in high-coverage regions. Finally, Nanopolish (v0.13.1) available online: https://c3g.github.io/covseq_McGill/SARS_CoV2_Sequencing/ONT_overview.html (accessed on 21 January 2024) was used to call mutations in regions with a minimum depth of 16× (8× per strand) and a flank of 10 bp. After masking regions with coverage below 20×, mutations called by nanopolish were integrated into the reference using bcftools (v1.9) available online: https://samtools.github.io/bcftools/ (accessed on 21 January 2024) to create a consensus sequence. In all cases, MN908947.3 available online: https://www.ncbi.nlm.nih.gov/nuccore/MN908947 (accessed on 21 January 2024) was used as a reference genome. A full description of both pipelines (v4.1.2) can be found in the following URL: https://genpipes.readthedocs.io/ (accessed on 21 January 2024).

### 3.3. Phylogenetic Analysis and Mutational Spectrum

The “Pangolin” network was used to identify the sequence lineage of the consensus sequences from Quebec (PangoLearn available online: https://github.com/cov-lineages/pangoLEARN (accessed on 9 November 2021, Pangolin version 1.2.93) [22], and all sequences characterized as being from the B.1.160 lineage were used to generate a distance tree. The phylogenetic trees were generated using Nextstrain viewer [23] using the default settings. The code for producing mutational graphs can be found here: https://github.com/HussinLab/covid19_mostefai2021_paper (accessed on 21 January 2024).

### 3.4. Intra-Host Analysis

The dataset for the intra-host analysis consists of sequences from one patient from the LSPQ and one patient described by Lee et al. [10] The intra-host mutational patterns were compared to our in-house intra-host mutation database based on 147,537 representative samples. Each library was trimmed using TrimGalore! v0.6.0 available online: https://github.com/FelixKrueger/TrimGalore?tab=readme-ov-file (accessed on 21 January 2024) and then mapped to the reference genome MN908947.3 using bwa-mem v.0.7.17 available online: https://bio-bwa.sourceforge.net/ (accessed on 21 January 2024). The remaining amplicon sequences were trimmed using iVar available online: https://andersen-lab.github.io/ivar/html/manualpage.html (accessed on 21 January 2024) with a hybrid amplicon definition file combining ARTIC v3, v4, and v4.1 designs. Primary reads were kept using Samtools v.1.15.1 available online: https://www.htslib.org/doc/ (accessed on 21 January 2024). iSNVs below 5% for Illumina and 10% for Nanopore that are not found at a higher frequency in at least one time point per patient are likely to be sequencing errors and were filtered out. Reads containing reference and alternative alleles for positions in the spike’s RBD were extracted from the BAM files using ctDNAtools [24]. The number of reads containing different combinations of alternative and reference alleles was then compiled to determine the frequencies of the possible haplotypes.

### 3.5. Protein Expression and Purification

FreeStyle 293F cells (Invitrogen, Waltham, MA, USA) were grown in FreeStyle 293F medium (Invitrogen, Waltham, MA, USA) to a density of 1 × 10^6^ cells/mL at 37 °C with 8% CO_2_ with regular agitation (150 rpm). Cells were transfected with a plasmid coding for SARS-CoV-2 S RBD using ExpiFectamine 293 transfection reagent, as directed by the manufacturer (Invitrogen, Waltham, MA, USA). One week later, cells were pelleted and discarded. Supernatants were filtered using a 0.22 µm filter (Thermo Fisher Scientific, Waltham, MA, USA). The recombinant RBD proteins were purified by nickel affinity columns, as directed by the manufacturer (Invitrogen, Waltham, MA, USA). The RBD preparations were dialyzed against PBS and stored in aliquots at −80 °C until use. To assess purity, recombinant proteins were loaded on SDS-PAGE gels and stained with Coomassie Blue.

### 3.6. Enzyme-Linked Immunosorbent Assay (ELISA)

The SARS-CoV-2 RBD ELISA assay used was previously described [16,20]. Recombinant SARS-CoV-2 S RBD protein (2.5 mg/mL), or bovine serum albumin (BSA) (2.5 mg/mL) as a negative control, were prepared in PBS and were adsorbed to plates (MaxiSorp Nunc, San Diego, CA, USA) overnight at 4 °C. Coated wells were subsequently blocked with blocking buffer (Tris-buffered saline (TBS) containing 0.1% Tween20 and 2% BSA) for 1 h at room temperature. Wells were then washed four times with a washing buffer (TBS containing 0.1% Tween20). CR3022 mAb (50 ng/mL) or a 1/250 dilution of plasma were prepared in a diluted solution of blocking buffer (0.1% BSA) and incubated with the RBD-coated wells for 90 min at room temperature. Plates were washed four times with a washing buffer followed by incubation with secondary Abs (diluted in a diluted solution of blocking buffer (0.4% BSA)) for 1 h at room temperature, followed by four washes. HRP enzyme activity was determined after the addition of a 1:1 mix of Western Lightning oxidizing and luminol reagents (PerkinElmer Life Sciences, Waltham, MA, USA). Light emission was measured with a LB942 TriStar luminometer (Berthold Technologies, Saint-Constant, CA, USA). The signal obtained with BSA was subtracted for each plasma and was then normalized to the signal obtained with CR3022 present in each plate. The seropositivity threshold was established using the following formula: mean of 10 pre-pandemic SARS-CoV-2 negative plasma + (3 standard deviation of the mean of 10 pre-pandemic SARS-CoV-2 negative plasma).

### 3.7. Cell Surface Staining and Flow Cytometry Analysis

293T cells were co-transfected with a GFP expressor (pIRES2-GFP, Clontech, Mountain View, CA, USA) in combination with plasmids encoding the full-length D614G or B.1.160 spikes. 48 h post-transfection, spike-expressing cells were stained with the CV3-25 mAb (5 ug/mL) or plasma (1/250 dilution). AlexaFluor-647-conjugated goat anti-human IgM + IgG + IgA Abs (1/800 dilution) were used as secondary Ab. The percentage of transfected cells (GFP + cells) was determined by gating the living cell population based on viability dye staining (Aqua Vivid, Invitrogen, Waltham, CA, USA). Samples were acquired on a LSRII cytometer (BD Biosciences, Franklin Lakes, NJ, USA), and a data analysis was performed using FlowJo v10.7.1 (Tree Star, Ashland, OR, USA). The seropositivity threshold was established using the following formula: (mean of 3 pre-pandemic SARS-CoV-2 negative plasma + (3 standard deviation of the mean of 3 pre-pandemic SARS-CoV-2 negative plasma). The CV3-25 mAb was used to normalize spike expression. CV3-25 was shown to effectively recognize all SARS-CoV-2 S variants [25].

### 3.8. ADCC Assay

This assay was previously described [25,26]. Parental CEM.NKr CCR5+ cells were mixed at a 1:1 ratio with CEM.NKr cells stably expressing a GFP-tagged full-length SARS-CoV-2 spike (CEM.NKr.SARS-CoV-2.S cells). These cells were stained for viability (AquaVivid; Thermo Fisher Scientific, Waltham, MA, USA) and with a cellular marker (cell proliferation dye eFluor670; Thermo Fisher Scientific, Waltham, MA, USA) to be used as target cells. Overnight-rested PBMCs were stained with another cellular marker (cell proliferation dye eFluor450; Thermo Fisher Scientific, Waltham, MA, USA) and used as effector cells. Stained target and effector cells were mixed at a ratio of 1:10. Plasma (1/500 dilution) was added to the appropriate wells. The plates were subsequently centrifuged for 1 min at 300 g and incubated at 37 °C and 5% CO_2_ for 5h before being fixed in 2% PFA. ADCC activity was calculated using the formula [(% of GFP+ cells in Targets plus Effectors) − (% of GFP+ cells in Targets plus Effectors plus plasma)]/(% of GFP+ cells in Targets) × 100 by gating on transduced live target cells. All samples were acquired using a LSRII cytometer (BD Biosciences, Franklin Lakes, NJ, USA), and a data analysis was performed using FlowJo v10.7.1 (Tree Star, Ashland, OR, USA). The seropositivity threshold was established using the following formula: (mean of 3 pre-pandemic SARS-CoV-2 negative plasma + (3 standard deviation of the mean of 3 pre-pandemic SARS-CoV-2 negative plasma)).

### 3.9. Virus Neutralization Assay

To produce pseudoviruses, 293T cells were transfected with the lentiviral vector pNL4.3 R-E- Luc (NIH AIDS Reagent Program) and a plasmid encoding for the indicated S glycoprotein (D614G or B.1.160) at a ratio of 10:1. Two days post-transfection, cell supernatants were harvested and stored at −80 °C until use. For the neutralization assay, 293T-ACE2 target cells were seeded at a density of 1 × 10^4^ cells/well in 96-well luminometer-compatible tissue culture plates (Perkin Elmer, Waltham, MA, USA) 24 h before infection. Pseudoviral particles were incubated with several plasma dilutions (1/50; 1/250; 1/1250; 1/6250; and 1/31250) for 1 h at 37 °C and were then added to the target cells followed by incubation for 48 h at 37 °C. Then, cells were lysed by the addition of 30 µL of passive lysis buffer (Promega, Madison, WI, USA) followed by one freeze–thaw cycle. An LB942 TriStar luminometer (Berthold Technologies, Saint-Constant, CA, USA) was used to measure the luciferase activity of each well after the addition of 100 µL of luciferin buffer (15 mM MgSO_4_, 15 mM KPO_4_ [pH 7.8], 1 mM ATP, and 1 mM dithiothreitol) and 50 µL of 1 mM d-luciferin potassium salt (Prolume, Randolph, MA, USA). The neutralization half-maximal inhibitory dilution (ID_50_) represents the plasma dilution required to inhibit 50% of the infection of 293T-ACE2 cells by pseudoviruses.

## 4. Results

### 4.1. Clinical Characteristics of the Patient

An immunosuppressed 73-year-old female with non-Hodgkin lymphoma first tested positive for SARS-CoV-2 (PANGO lineage B.1.160) on the 8 of January 2021 (Day 1 (D1) referring to the day of the first positive test). She had undergone several courses of anti-CD20 (rituximab) and chemotherapy in the months preceding her COVID-19 diagnosis. She was vaccinated with the Pfizer-BioNTech vaccine on the 25 of February 2021. After a negative test on the 31 of March 2021, the patient tested positive again on the 28 April 2021 (D111) (Appendix A). The full timeline of her infection is shown in Figure 1a. Because the sample sequenced on D111 had S:E484K, it was first assumed that this sample and all subsequent time points were from a reinfection. However, a phylogenetic analysis of all time points shows that all samples came from the same initial infection that lasted at least 173 days, from the 8 of January to the 29 of June 2021 (Figure 1b). She then tested negative two days in a row on the 5 and 6 July 2021 before passing away in August of 2021 from a non-COVID related complication. She was not treated with mAbs or convalescent plasma. The complete list of samples, dates, and tissues can be found in Appendix A.

### 4.2. Intra-Host Analysis of Four Sequential Samples

At D1, the patient had all the characteristic mutations of B.1.160 in Quebec, as well as eight additional mutations (Figure 2) shared with five other sequences in the LSPQ database (Figure 1b). These eight mutations were only seen together in B.1.160 within Quebec, making this sub-variant Quebec-specific. The B.1.160 characteristic mutations, as well as all the Quebec-specific mutations, were all still present at the last time point examined (D172). Sequences at D111 and D116 shared ten new mutations that were not seen at D1 nor at D172. The reversal of all 10 consensus mutations acquired at D111/D116 made it highly unlikely that the substrain at D172 evolved from the ones at D111/D116. Additionally, the substrain observed at D172 accumulated 20 new mutations that were not shared with other time points. The only notable exception was C26728T, a synonymous mutation in the N protein present at frequencies of 0.29, 0.2, and 0.92 on D111, D116, and D172, respectively; the most parsimonious explanation for this pattern is that C26728T is a recurrent mutation. Thus, D111/D116 and D172, sampled 61 days apart, show two different lineages descending from D1, both of which are seen in the nasopharyngeal tissue. None of the D111/D116-specific nor D172-specific mutations show intra-host heterogeneity, meaning an intra-host frequency lower than 95% at these positions. This suggests that the D172 substrain (or an ancestor of the D172 substrain) was not present in the same pool as the D111/D116 substrain. Given that this variant was not circulating at that time in Quebec, this result suggests that the D172′s substrain likely evolved in another anatomical location in the patient’s body, where it possibly accumulated all the mutations seen at D172. This substrain was able to reinfect the nasopharyngeal tissue after the D111/D116 substrain was cleared.

### 4.3. Intra-Host Evidence of Multiple Viral Populations with Distinct Spike RBD Mutations

We performed a detailed analysis of the co-occurrence of mutations in the intra-host data and observed that the substitutions on S:484 at D111 and D116 were mutually exclusive; no reads contained both alternative alleles at positions 23012 and 23013 (Appendix A). There were two major distinct mutant populations of S:E484K (0.68 on D111 and 0.93 on D116) and S:E484A (0.26 on D111 and 0.06 on D116), as well as a small wild-type population (0.06 on D111 and 0.01 on D116).

We looked for other case studies of immunosuppressed patients infected during that time frame with raw sequencing data available. A study by Lee et al. [10] described two leukemia patients with protracted SARS-CoV-2 infections, one of which (designated as P2) had evidence of multiple intra-host viral populations. P2 had G22599A and C22605T on D38 which were both at a frequency of 0.10 (Appendix A), potentially suggesting the co-occurrence of these mutations. However, when retrieving reads containing both positions, we see that those substitutions belong to different viral populations (Appendix A), highlighting the importance of analyzing aligned reads to describe the intra-host population dynamics. Like our patient, P2 had substitutions G23012A and A23013C on S:484. P2 also had two distinct viral populations with different mutations on S:346 at a single time point.

We then looked for intra-host evidence of multiple viral populations for S:346 and S:484 in the general population from samples collected during the years 2020 and 2021. An analysis of iSNVs in 147,537 SARS-CoV-2 sequencing libraries downloaded from the NCBI revealed that no sequence had more than one mutation on codon S:346. Only four samples had more than one mutation on S:484 that led to distinct viral populations (SRR15258550, SRR15061404, SRR17006835, and SRR16298333). No clinical details on these infections are available, but the overall mutational burden was not characteristic of protracted infections. The small number of occurrences of the RBD mutational pattern found in the general population during that time (at a frequency below 0.003%) highlights the peculiar character of the mutational events identified in the immunosuppressed individuals described here.

### 4.4. Analysis of Patient’s Plasma

To test whether the patient mounted an antibody response to the Pfizer-BioNTech vaccine they received, we analyzed the available patient’s plasma collected between 5 May 2021 and 10 May 2021. Despite having been infected with SARS-CoV-2 for almost four months and having received one dose of the vaccine, the patient developed a weak immune response to the virus (Table 1). Specifically, the total immunoglobulin (Ig) and immunoglobulin G (IgG) levels were found to be lower than typically observed in a cohort of healthcare workers who were previously infected and received one dose of the vaccine [16]. Consistent with the low level of antibodies, the patient plasma presented low Fc-effector functions as measured with an antibody-dependent cellular cytotoxicity (ADCC) assay (Appendix A) and no neutralizing activity against pseudoviruses bearing the following spikes: S:D614G or S:S477N + S:D614G.

## 5. Discussion

We performed an intra-host analysis on serial SARS-CoV-2 sequences from a patient with non-Hodgkin lymphoma and compared the identified patterns with 147,537 sequencing libraries to look for intra-host populations of immune escaping mutations in the spike’s RBD. Consistent with the literature, we found evidence of multiple viral populations co-existing in this region within a single immunosuppressed patient [27]. Furthermore, we found distinct populations of mutants for codon S:484 which was extremely rare in the general population during the first two waves of the pandemic. The relatively weak antibody response in this patient after D116, in line with recently reported findings [28,29], suggests the possibility of the emergence of RBD mutations under incomplete immune pressure post-vaccination, leading to the dominance of a different viral quasispecies as this pressure waned. Several studies looking at intra-host mutational patterns in immunosuppressed patients have found that mutations favoring immune escape that commonly arise in these infections can become fixed in VOIs and VOCs months later [27]. Thus, this stresses the importance of studying this population in a longitudinal design to obtain insights into key steps of viral evolution. Furthermore, we advocate for the publication of sequencing reads along with the publication of consensus sequences to facilitate intra-host analyses.

When comparing consensus mutations found in our patient’s D1, D111/D116, and D172 samples, we saw that all four time points share the mutations present at D1, demonstrating that this is a single long-lasting infection. However, substrains at D111/D116 and D172 accumulated 10 and 20 new mutations, respectively, that are not shared across time points, implying that the substrains present at these time points evolved from the substrain at D1 independently of each other. The substrain present at D111/D116 was cleared from the nasopharyngeal tissues and was later replaced by the one found at D172. B.1.160 and its sub-variants were no longer circulating in the Quebec population by April 2021 (LSPQ surveillance data), so the likelihood that the patient was reinfected by the same sub-variant from January is extremely low. Given the lack of overlap of novel mutations between D111/D116 and D172, the substrain present at the last time point is very likely to have evolved in another anatomical location, which is consistent with patterns observed in viral reservoirs [9].

Previous work has shown how SARS-CoV-2 can persist in different anatomical locations during infection, as well as months after the infection has been cleared from the respiratory system, accumulating mutations leading to new SARS-CoV-2 sequences. Recent evidence, including a study involving a patient with non-Hodgkin lymphoma [13], has highlighted the potential for novel substrains possibly from viral reservoirs, originating from initial infecting strains, to reinfect and replicate in the respiratory tract. Our intra-host phylogenetic analysis contributes to this growing body of evidence, emphasizing the role of viral reservoirs in the emergence of SARS-CoV-2 variants from prolonged infections in immunocompromised individuals. Given that the sampling of viral sequences was performed in only one anatomical location, our study is, however, not able to determine which organ is the source of the reservoir. The gut has been proposed as one assumed location for viral reservoirs of SARS-CoV-2 [30,31]. Other studies have reported on the neuroinvasive capacity of SARS-CoV-2 [32], which is difficult to confirm in patients during an infection due to lack of technology to directly sample central nervous system tissues. Nevertheless, local and longitudinal measurements of viral proteins across organs, as well as detailed imaging studies [33], could help identify the location of viral reservoirs in future studies. Finally, the theory of viral reservoirs as an explanation for long-COVID symptoms has been put forward because viral antigens or intermediate molecules of viral replication have been detected in long-COVID cases despite negative PCR tests [34,35,36,37]. These results thus call for further investigation to determine whether SARS-CoV-2 viruses present in viral reservoirs have the capacity to reinfect immunocompetent patients [9].

As we approach the fourth year of SARS-CoV-2 transmission in humans, we see that the current circulating strains have accumulated many mutations in the RBD of the spike glycoprotein that were first documented in immunosuppressed patients in earlier stages of the pandemic, such as S:484 described here which became widespread with the arrival of the Omicron variant [38]. These recurrent mutations can fluctuate during these protracted infections and may or may not reach fixation [11]. It is still unclear whether SARS-CoV-2 evolution in the general population is driven by protracted infections as there is little evidence of an outbreak being traced back to an immunosuppressed patient, with the potential exception of Alpha [39]. Nevertheless, in the case of a future viral outbreak, genomic surveillance of the viral populations through regular sampling of immunosuppressed patients early into the pandemic would provide insight into potential viral evolutionary paths.

## Figures and Tables

**Figure 1 viruses-16-00342-f001:**
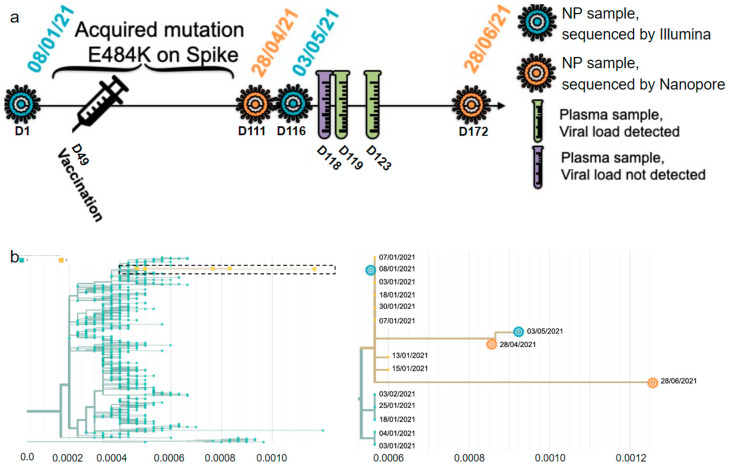
Description of the infection. (**a**) Timeline of the infection and type of sample per date. NP, nasopharyngeal. (**b**) Distance tree of lineage B.1.160 in Quebec on the left, and close up of the box containing the patient’s four consensus sequences on the right. The *X* axis is measured in substitutions per site per year.

**Figure 2 viruses-16-00342-f002:**
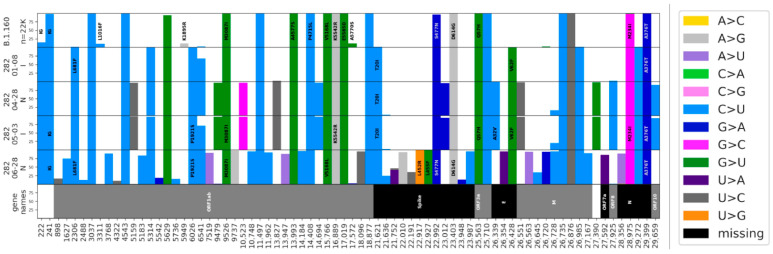
Allelic frequencies in B.1.160 in Quebec and in the patient’s infection. The top row displays the percentages for a total of 2627 B.1.160 consensus sequences from the LSPQ database. The four following rows show intra-host frequencies for the patient’s mutations for each time point. Only mutations with intra-host frequencies above 5% for Illumina sequences (D1 and D116) and 10% for Nanopore sequences (D111 and D172) for at least one time point are presented. Because of the respective error rates of both sequencing technologies, discrepancies of up to 5% for Illumina sequences (D1 and D116) and 10% for Nanopore sequences (D111 and D172) are likely to be sequencing artifacts. Non-synonymous mutations are written on top, and the color represents the nucleotide change. We observed a genome-wide total of 22 C to U substitutions on 55 intra-host mutations (40%). The additional mutations seen on D111 in Figure 1b but not in Figure 2 can be explained by the different filters used in the default Nextstrain settings and our in-house pipeline (detailed in the Section 3).

**Table 1 viruses-16-00342-t001:** Anti-RBD Ig levels and neutralization capacity.

Sample	ELISA Anti-RBD (RLU Normalized to CR3022)				Neutralization (ID_50_)	
	IgG	IgM	IgA	Total Ig	D614G	B.1.160 (S477N + D614G)
D118	17.242	0.234	0.892	14.079	30	30
D119	11.042	0.317	1.011	10.771	30	30
D123	8.835	0.409	1.051	11.509	30	30
Threshold	4.834	3.216	0.900	7.036	30	30

## Data Availability

The raw reads can be found via SRA under Bioproject PRJNA1034519.

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
