# Peer review of "Intra-Host Evolution Analyses in an Immunosuppressed Patient Supports SARS-CoV-2 Viral Reservoir Hypothesis"

_viruses, 2024, doi:10.3390/v16030342_

Round 1
Reviewer 1 Report
Comments and Suggestions for Authors
This manuscript presents the results of an in-depth analysis of a single immunosuppressed patient suffering from non-Hodgkin lymphoma and an infection with SARS-CoV-2. This is a fascinating report, also because the multifacetted study brings to light very interesting aspects of SARS-CoV-2 genetics. It is true that the data have been derived under immunosuppression from a single patient. Nevertheless, new aspects of SARS-CoV-2 infection and its genetic have been discovered:
The virus has persisted in this patient over several months.
The authors have performed in vivo studies on viral mutants, particularly affecting the receptor binding domain of the viral spike protein. Among other interesting details, the authors document the persistence and evolution of different viral mutants in different organ systems.
There are only two items, I wish to address that will possibly be of interest to the authors.
(i) It is stricking that C to U mutations seem to be rather rare. Previous studies on SARS-CoV-2 mutants by several groups have detected a relatively high proportion of C to U transitions, even up to 40%. This did not seem to be the case in this study. Could immune suppression somehow be involved?
(ii) The authors may be interested in a genetic study of the problem on purely theoretical grounds:
Weber et al. 2022 in the same journal . Viruses 2022, 14, 885. https://doi.org/10.3390/v14050885. This report concludes:
"It will take systematic investigations to explore the possibility of whether these mechanisms (patch type sequence identities among a huge number of very diverse species) might have served also in the generation and selection of variants of newly emerging pathogenic viruses like SARS-CoV-2 and others." And there is more to come on this topic.
Author Response
Reviewer 1
This manuscript presents the results of an in-depth analysis of a single immunosuppressed patient suffering from non-Hodgkin lymphoma and an infection with SARS-CoV-2. This is a fascinating report, also because the multifacetted study brings to light very interesting aspects of SARS-CoV-2 genetics. It is true that the data have been derived under immunosuppression from a single patient. Nevertheless, new aspects of SARS-CoV-2 infection and its genetic have been discovered:
The virus has persisted in this patient over several months.
The authors have performed in vivo studies on viral mutants, particularly affecting the receptor binding domain of the viral spike protein. Among other interesting details, the authors document the persistence and evolution of different viral mutants in different organ systems.There are only two items, I wish to address that will possibly be of interest to the authors.
(i) It is stricking that C to U mutations seem to be rather rare. Previous studies on SARS-CoV-2 mutants by several groups have detected a relatively high proportion of C to U transitions, even up to 40%. This did not seem to be the case in this study. Could immune suppression somehow be involved?
> We thank the reviewer for their positive comments and for dedicating their time to review our manuscript.
Regarding the comment about the C to U mutations, we observed that 22 out of 55 mutations are C to U transitions (as seen in Figure 2), aligning with the expected 40% rate genome-wide. However, the reviewer is right to note that this general rate contrasts on patterns specific to the Spike protein, where C to U mutations are found to be less frequent. This discrepancy is also highlighted in Figure S1, which focuses solely on the Spike region. The distinction between the Spike protein and the rest of the genome underscores the unique selective pressures on the Spike, but a global statement about these cannot be concluded from one patient.
In response to your comment, we added a clarification to Figure 2 legend:
“Genome-wide, we observed a total of 22 C to U substitutions out of 55 intra-host mutations (40%)”
(ii) The authors may be interested in a genetic study of the problem on purely theoretical grounds:
Weber et al. 2022 in the same journal . Viruses 2022, 14, 885. https://doi.org/10.3390/v14050885. This report concludes:
“It will take systematic investigations to explore the possibility of whether these mechanisms (patch type sequence identities among a huge number of very diverse species) might have served also in the generation and selection of variants of newly emerging pathogenic viruses like SARS-CoV-2 and others.” And there is more to come on this topic.
> We appreciate the reference to Weber et al. 2022 and the highlighted insights on the genetic study of patch type sequence identities. We acknowledge the importance of this area for understanding the evolution of emerging viruses. While our current study does not directly explore this aspect, we find the suggestion valuable and will consider this line of investigation in our future research.
Reviewer 2 Report
Comments and Suggestions for Authors
Dear authors,
This is a well written paper, however, it is a bit outdated research. I suggest to add more in discussion ( and may be add in the introduction) about the known and published cases of intrahost evolution of SARS-CoV-2 in immunocompromised patients. Also, there have been papers also discussing that these patients typically do not have pronounced antibody response to the virus and that viral mutations may accumulate in the absence of humoral immune response. I suggest that you include thiese aspects into your discussion section.
Author Response
This is a well written paper, however, it is a bit outdated research. I suggest to add more in discussion ( and may be add in the introduction) about the known and published cases of intrahost evolution of SARS-CoV-2 in immunocompromised patients. Also, there have been papers also discussing that these patients typically do not have pronounced antibody response to the virus and that viral mutations may accumulate in the absence of humoral immune response. I suggest that you include thiese aspects into your discussion section.
> We thank the reviewer for their feedback and valuable suggestions. We have made three key adjustments to our manuscript to ensure the timeliness of research and to comprehensively address recent developments in the study of SARS-CoV-2, especially regarding immunocompromised patients.
Firstly, in the Introduction, we updated the text to reflect the availability of longitudinal datasets on SARS-CoV-2 evolution in these patients
“Recent studies have begun to fill this gap, providing longitudinal datasets of sequences collected from these individuals at different time points during their infections (11-13)”,
which replaced:
“Unfortunately, few longitudinal datasets of sequences collected from these individuals at different time points during their infections are available.”
Secondly, in the Discussion, we incorporated a reference to recent findings on the weak antibody responses and how this may influence the emergence of viral mutations. We added:
“The relatively weak antibody response in this patient after D116, in line with recently reported findings (18,19), suggests the possibility of the emergence of RBD mutations under incomplete immune pressure post-vaccination, leading to the dominance of a different viral quasispecies as this pressure waned.”
Lastly, we revised our discussion to incorporate a recent study of a patient with the same condition as our patient, which corroborates our results. We changed:
“However, to our knowledge no study has shown examples of novel substrains from another location acquiring the capacity of re-infecting and replicating in the respiratory tract. As such, our intra host phylogenetics analysis shows how a putative viral reservoir in an immunosuppressed patient has the capacity to generate new substrains able to infect the nasopharyngeal tissue”
to
“Recent evidence, including a study involving a patient with non-Hodgkin lymphoma (13), has highlighted the potential for novel substrains possibly from viral reservoirs, originating from initial infecting strains, to reinfect and replicate in the respiratory tract. Our intra-host phylogenetics analysis contributes to this growing body of evidence, emphasizing the role of viral reservoirs in the emergence of SARS-CoV-2 variants from prolonged infections in immunocompromised individuals.”
Additional references:
11. Chaguza C, Hahn AM, Petrone ME, Zhou S, Ferguson D, Breban MI, et al. Accelerated SARS-CoV-2 intrahost evolution leading to distinct genotypes during chronic infection. Cell Rep Med [Internet]. 2023 Feb 21 [cited 2023 Sep 27];4(2). Available from: https://www.cell.com/cell-reports-medicine/abstract/S2666-3791(23)00035-6
12. Brandolini M, Zannoli S, Gatti G, Arfilli V, Cricca M, Dirani G, et al. Viral Population Heterogeneity and Fluctuating Mutational Pattern during a Persistent SARS-CoV-2 Infection in an Immunocompromised Patient. Viruses. 2023 Feb;15(2):291.
13. Villaseñor-Echavarri R, Gomez-Romero L, Martin-Onraet A, Herrera LA, Escobar-Arrazola MA, Ramirez-Vega OA, et al. SARS-CoV-2 Genome Variations in Viral Shedding of an Immunocompromised Patient with Non-Hodgkin’s Lymphoma. Viruses. 2023 Feb;15(2):377.
18. Hettle D, Hutchings S, Muir P, Moran E. Persistent SARS-CoV-2 infection in immunocompromised patients facilitates rapid viral evolution: Retrospective cohort study and literature review. Clin Infect Pract. 2022 Nov 1;16:100210.
19. Li Y, Choudhary MC, Regan J, Boucau J, Nathan A, Speidel T, et al. SARS-CoV-2 viral clearance and evolution varies by type and severity of immunodeficiency. Sci Transl Med. 2024 Jan 24;16(731):eadk1599.